# Virtual field experiences in a web-based videogame environment: Open-ended examples of existing and fictional field sites

Mattathias D. Needle[1], Juliet G. Crider[1], Jacky Mooc[2], and John F. Akers[2]

[1]Department of Earth & Space Sciences, University of Washington, Seattle, USA
[2]UW Reality Lab, Department of Computer Science and Engineering, University of Washington, Seattle, USA

*Correspondence to*: Mattathias D. Needle (mneedle@uw.edu)

**Abstract.** We present two original, videogame-style field-geology experiences designed to allow flexible, open-ended exploration for geologic mapping and structural geology. One simulation features the Whaleback anticline, a site in central Pennsylvania (USA) with three-dimensional exposure of a 30-m-high fold, based on a terrain model that was acquired through structure-from-motion photogrammetry. The second example is a fictional location with simplified geology, built with digital modeling software and inspired by the geology of northwestern Washington. Users move through the terrain, as if in the field, selecting where to make observations of the geologic structure. Additionally, these virtual field experiences provide novel visualization opportunities through tools like a geodetic compass that instantly plots data to a stereonet, and a jetpack simulation which allows the user to interrogate geologic surfaces in hard-to-reach locations. We designed the virtual field experiences in a widely used videogame-creation software and published the field simulations for access via the internet and common web browsers, so that no special hardware or software is required to play. We implemented these field simulations to partially replace field and lab exercises in two different courses offered remotely through the University of Washington Department of Earth and Space Sciences, with assignments that address many of the learning goals of traditional in-person exercises. Because the virtual field experiences are open ended, other instructors can design different exercises to meet different learning goals. While this game environment currently serves as an enhancement to remote education, this format can also augment traditional educational experiences, overcoming several challenges to accessing the field or particular outcrops and thereby broadening opportunities for participation and scientific collaboration.

## 1 Introduction

Starting in March 2020, safety measures to dampen the transmission of COVID-19 impacted field instruction related to geoscience education. While the COVID-19 pandemic forced geoscience educators to design alternatives to in-person field instruction, this demand for remote instruction also highlighted shortcomings of the traditional geoscience curriculum to include students who for a multitude of reasons have difficulties accessing traditional field-based coursework (e.g., Wolfe and Riggs 2017; Carabajal, Marshall, and Atchison 2017). The need for educational experiences that incorporate fundamentals

traditionally taught in field-based courses will remain after the pandemic. Such educational experiences might be deployed for the purpose of inclusion or for skill development in data collection and analysis as an alternative or precursor to traditional field courses.

35 Historically, the emergence of geoscience in Europe and North America was closely tied to objective descriptions of outcrops and discussion of the observations in the field (e.g. Hallam 1990). The practice dates from at least the time of James Hutton in the late 1780's (e.g. Gould 1982), and the tradition continues in modern field conferences (e.g. Evenson et al. 2000) and field trips associated with professional geoscience assemblies (e.g. Lageson et al. 1999). Contemporary geology education recapitulates this process, with Bachelor degrees commonly culminating in a required capstone field course (Whitmeyer et al. 40 2009).

In the Department of Earth and Space Sciences at the University of Washington, capstone field instruction includes components of objective rock description at outcrops and measurement of bedding orientation and other structural data for the purpose of mapping, constructing cross-sectional interpretations from maps, and interpreting a geologic history from the observed data. 45 A course typically taken in the first year of the major also includes these components in a staged, fictional scenario with hand samples in the classroom. Remote-learning mandates related to the COVID-19 pandemic precluded field or classroom instruction, and we sought alternative approaches that could address at least some of the original learning goals of these exercises.

50 We designed video-game-style exercises to provide a three-dimensional, first-person-perspective, virtual field experience accessible via standard internet browsers. Here, we present two examples of virtual field experiences that we implemented during the COVID-19 pandemic for remote instruction in: 1) the department's capstone field-course as part of a module on folding; and 2) an introductory geology course as a substitute for the usual in-classroom rock-identification and map-making final laboratory project. Each virtual field experience incorporates a terrain model that was generated through different means: 55 a structure-from-motion (SfM) model made from drone-captured photographs for the capstone course; and a fictional terrain model designed in 3D-modeling software for the first-year geology course. Here, we report on pandemic-related "emergency" replacements for field exercises, with a prototype game to provide proof-of-concept for our approach.

**2 Overview of Virtual Field Experiences**

We aimed to make an experience in which students could explore outcrops and terrains in first-person-perspective without 60 limitations on where they could collect data. We designed the virtual field experiences in Unity (Unity Technologies 2020), a cross-platform game engine in which we imported our geologic terrain models into a 3D environment and wrote scripts to govern how the users (in this case, geoscience students) can interact with the outcrops and terrain. The cross-platform nature

of Unity allowed us to choose how to package the virtual field experience for student access. We chose WebGL, a JavaScript application-programming interface (API) for rendering interactive 3D graphics within web browsers, as the ideal platform for sharing the virtual field experiences. By building the exercises for the web, anyone with a computer and internet connection can be granted access, no special equipment (e.g., headset, glasses, or joystick) is needed, and no extra steps in downloading an operating-system-specific application are required.

Here, we report on the some of the functions within our virtual field exercises, but we intend to publish a detailed report on how we developed our virtual-field-experience interface in Unity. Our workflow and Unity-based software package for generating a video-game-style virtual field experience is currently available at https://github.com/UWRealityLab/StructuralQueryToolkit so that anyone with a 3D model can design their own virtual field experience and leverage the tools we developed for education and/or research.

The user interface for our course-related virtual field experiences simulates some classic geology tools with novel visualization abilities: A distance-measuring tool permits linear measurements between multiple points in 3D space and prints distances along user-generated line segments. Toggling the map view grants an orthogonal aerial view of the terrain and shows the user's position. The video-game setting also permits tools that are not typically available to geologists: A jetpack allows users to fly over the terrain and enables interrogation of outcrops that would typically be difficult or unsafe locations at which to collect data (Fig. 1a). The user can control movement in all directions with their computer-keyboard arrow keys and change perspective using their mouse or track pad. While in jetpack mode, the user can also change elevation.

Our Unity-based stereonet tool (written in C# programming language) is a novel way to collect data from surfaces and instantly have the data printed to a stereonet within the user's view. When the stereonet tool is activated, the user can click on surfaces within the virtual field experience and take three different types of measurements: 1) planes (Figs. 1b-c), for which the user generates polygons on the surface; 2) poles to planes (Fig. 1d), for which the user clicks once on the surface and generates a flag normal to the surface; and 3) lineations (Fig. 1e), for which the user clicks twice to generate a line on the surface. For each measurement, the data is not only plotted to the stereonet, but the numeric values (strike-and-dip, trend-and-plunge) are printed on the screen (e.g., Fig. 1a). The pole component has two extra features: pole-style measurements also print the elevation at the measured point; and once several poles are collected, the user can initiate a stereonet pi-plot (e.g., Marshak and Mitra 1988, p. 157) which indicates an approximate trend and plunge of a fold axis (Fig. 1a) and updates as more measurements are taken. Data acquired with these tools can be exported for use in Strabo (Walker et al. 2019), Stereonet (Cardozo and Allmendinger 2013) and other plotting and analysis software.

## 3 The Whaleback Anticline Virtual Field Experience

95      As part of the remote instruction for our department's capstone summer field-geology course, we devised a module on fold geometry that included a virtual field trip. Creation of the virtual field experience leveraged an existing SfM-derived terrain model of folded sedimentary rocks. The assignment related to this virtual field experience revolved around the special opportunities to study folds at this site in the field, while also providing the opportunity to collect data from locations at the field site that are not accessible on foot.

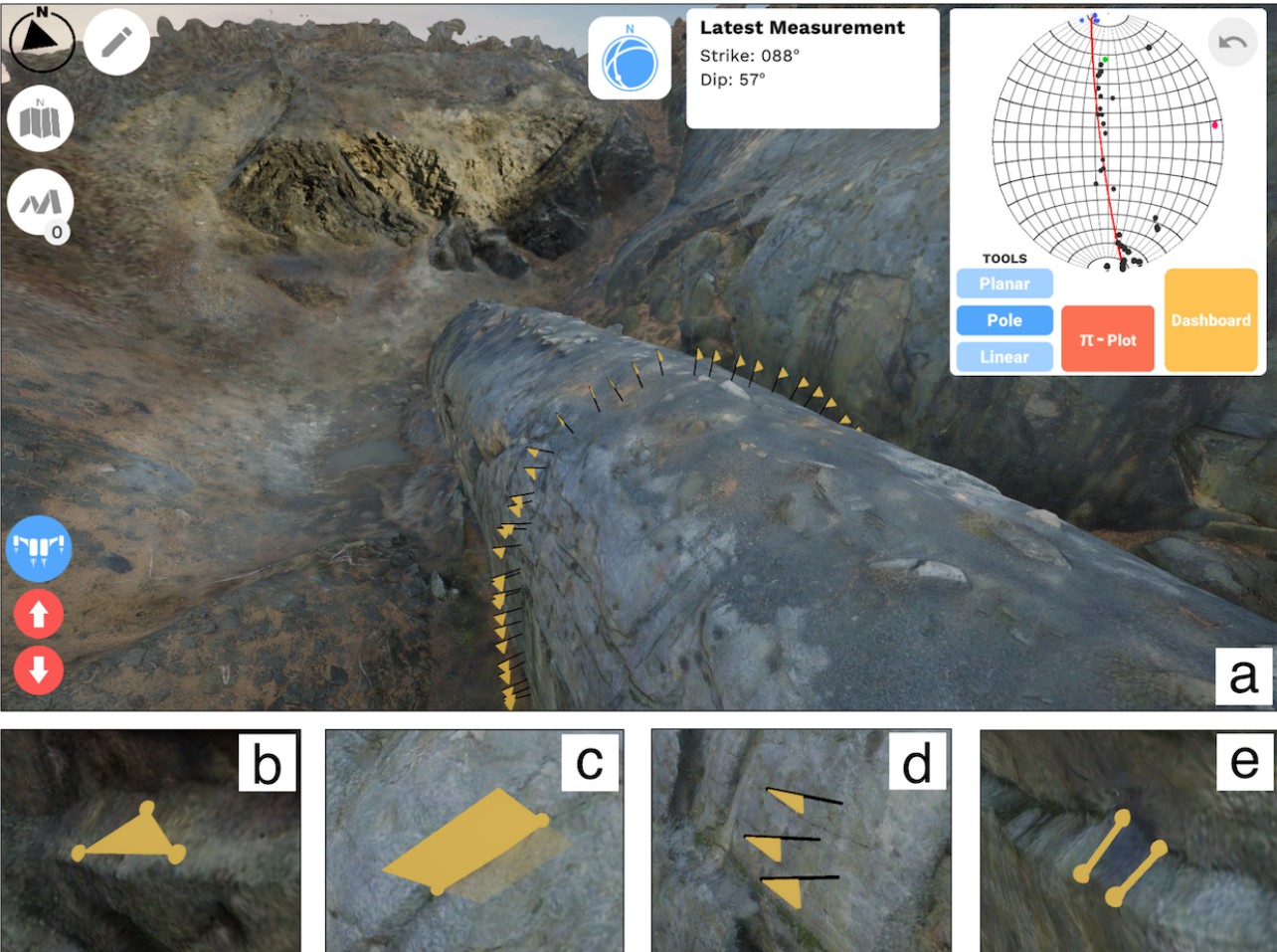

100

**Figure 1:** The interface and custom geology tools within our virtual field experiences. (a) View of the Whaleback Anticline while the user is in jetpack mode. From the top right and continuing counter-clockwise, icons on the perimeter enable the user to toggle the stereonet measurement tools, drawing tools, a compass, map (orthographic aerial) view, linear-distance measurement tool, and jetpack mode. Poles to bedding, represented by orange flags, are plotted as a scanline roughly perpendicular to the trend of the fold crest. The stereonet pi-plot, 105 shows the measured poles and automatically updates as more data is collected. A best-fit great circle and corresponding pole (red line and point on stereonet) calculate the approximate trend and plunge of the fold axis. (b-e) User-constructed representations of measurements include 3-point planes, 2-point-and-rotate planes, poles to planes (flags), and lineations.

The Bear Valley Strip Mine (Shamokin, Pennsylvania) is a popular destination for structural-geology field trips due to its excellent, three-dimensional exposure of an excavated sandstone fold train, of which the central anticline is named "The Whaleback" (Fig. 1a; Nickelsen 1979; Levine and Eggleston 1992). Visitors can walk along the crests of two anticlines, or next to the 200-m-long fold limbs, which feature fossils, concretions, joints, and decimeter-scale secondary faults. Because the Whaleback is 30 m high (from trough to crest) with steeply-dipping/overturned limbs, most of the sandstone surface is inaccessible to direct measurement in the field. To study the geometry of the Whaleback for a research project, drone-acquired photographs of the strip mine were used with SfM photogrammetry to make a 3D point cloud of the surface in AgiSoft PhotoScan Professional (2018). We edited the point cloud by removing as much vegetation as possible to preserve only the exposed sandstone surface and sediment cover. Within the PhotoScan workflow, a polygonal mesh was generated from the point cloud, along with an associated texture which contains the color information for the mesh. Because the photogrammetry utilized real-world coordinates, the polygonal mesh (subsequently referred to as a "model") is scaled to real coordinates (UTM and elevation) and dimensions (meters). This model forms the basis of the virtual field experience; our virtual geology tools (compass and stereonet, ruler, map, and jetpack) provide the mechanism for the students to explore the virtual field site and collect data. In addition to the 3D model, we also produced an orthorectified aerial image of the site to use as a base map (Fig. 2c).

We piloted this virtual field experience in July 2020 with 31 undergraduate geology students enrolled in University of Washington's remote field course, in a day-long workshop. Following a refresher on stereonets and fold geometry, the students received access to the Whaleback WebGL-based virtual field experience which was hosted on a department server. The assignment had four components: First, to familiarize them with the game controls, we asked the students to explore the strip mine by walking or flying(!), screen-capture an interesting geologic feature, and write an associated objective description. Many students noted the anticline and syncline on the mine's eastern wall (Fig. 2a; the syncline is apparent in the background of Fig. 1a); however, one student wrote an interesting geological description of some automobile tires that are remnants of trash within the mine. To help students understand the geologic compass function and practice the transition from 3D "field" perspective to 2D map, the students took strike-and-dip measurements of the surface of the Whaleback and plotted strike-and-dip symbols on the orthorectified aerial image. Once they were fully introduced to the game, we asked the students to investigate the along-trend variability of the Whaleback Anticline. Students used the pole tool to create transects across the fold in four different areas that they selected (e.g., orange flags plotted in Fig. 1a). They then produced annotated aerial images, with four stereonets to show how the trend-and-plunge of the fold axis varies longitudinally (Figs. 2b-c). From their data, students were able to observe that the Whaleback is doubly-plunging rather than purely cylindrical. The fourth exercise required constructing a profile of the fold train and interpolating the synclines that presumably connect the three exposed anticlines. In order to make this profile, students used the "pole" tool to measure elevation on the surface and the distance-measuring tool to constrain horizontal distance a long a transect perpendicular to the fold, and plotted the values, interpolating below the sedimentary cover.

Implementation of the exercise was straightforward. Regarding the WebGL-performance on students' machines, no computing

145 or model-rendering problems were reported. Although students were not discouraged from working in groups, the students

primarily worked alone for these exercises.

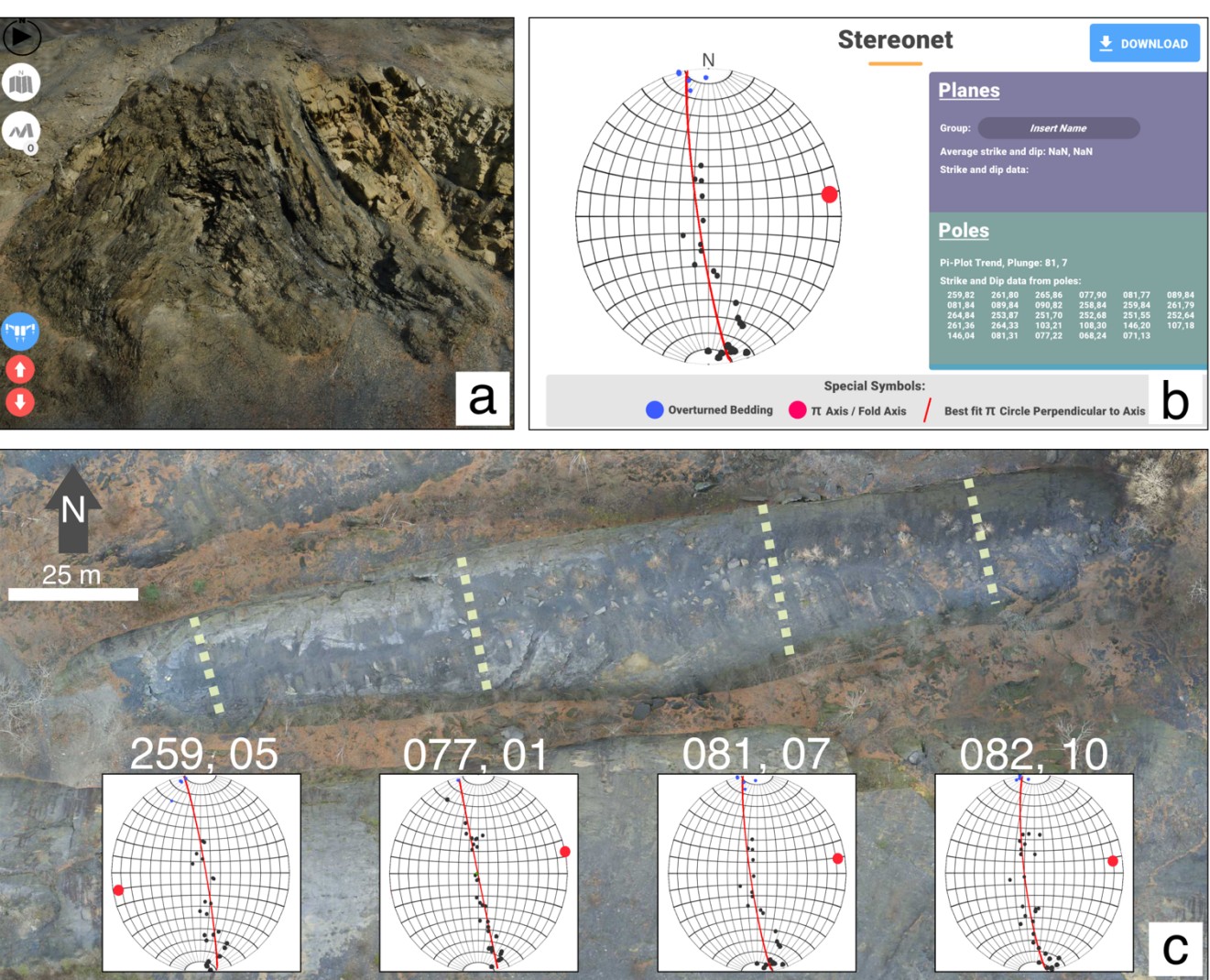

**Figure 2:** Examples of exercises and data. (a) An anticline (and often, the adjacent syncline) exposed in profile in the east wall of the strip
150 mine was a common feature that the students selected for description. The field of view is 55 meters wide. (b) Screenshot of data from a
scanline used to determine the trend and plunge of the fold axis of the Whaleback in the central eastern quarter. The fold axis orientation,
generated from an automatic pi-plot feature, is exaggerated as a red circle; the data for each measurement is printed on the right. (c) Students
annotated an orthophoto of the Whaleback with stereonet data to investigate variations in the fold axis and best characterize the fold
geometry. Each stereonet is associated with a student-made scanline (shown as a dashed line) on a section of the fold, with the trend-and-
155 plunge values of the calculated fold-axis printed above.

The virtual experience we created is open-ended, as there are no specific checkpoints or pieces of information that the user is required to collect; rather, the user chooses what data to collect and where to make those observations. Because of flexibility this virtual field experience, other instructors can design exercises appropriate to their courses and students. In addition to the exercises we piloted, students could: make a structure contour map of the top of the Whaleback sandstone using the elevation tool; compare arc-length to wavelength of the fold; examine meso-scale structures on the larger fold; compare pi-diagram (poles to bedding) to beta-diagram (intersecting great circles) to direct measurements of the fold hinge line for representation of the fold axis; compare axes and shapes of different folds (Whaleback vs. North anticline); compare folds in different stratigraphic horizons (Whaleback vs. syncline above); and there are undoubtably other possibilities. In short, the open structure of the virtual field experience gives instructors similar opportunities that they have in the field to adapt to the needs of their students.

At the time of the pilot exercise, we gave the students an optional, open-ended opportunity to provide feedback about the game: ("If you have any comments regarding improvements to the interface in the Whaleback part of the game, please let me know in the space below!") Although the prompt requested feedback regarding improvements to the game interface, we also received comments related to the student learning experience. Note that, although responses to this question were optional, it was included as part of a graded assignment, and this may have affected how students responded. Out of 31 students, fourteen students (48%) noted that the virtual experience was aesthetically pleasing and/or fun. Examples of these statements are "I thought it was great! It's a really cool way to visualize an area we are not able to visit," and "This is a lovely game! I absolutely enjoyed it and it really helped with the immersion. It was super enjoyable!" On the other hand, one student reported getting "scared by the walking sounds because it's the same soundbite a lot of pixel horror games use." Positive remarks about the interface were made by eleven students (35%). Examples of these are "I really liked the interface; I thought it was very user-friendly and intuitive to use.", "I think that the games works very well and is pretty intuitive", and "I really liked the map mode button which created a standardized view of the map; it made it easy to orient myself and compare my map view to the instructions." An additional eight students (26%) offered only suggested improvements to the interface. Examples of these include, "I think an F1, instructions command that provides the full scope of the program would be helpful!", and "Add an ability to click/highlight a previous point so that we can see what the strike and dip values were. This way, we don't have to go back and redo it to find out what the values were." Finally, six students (19%) volunteered that their education was enhanced by the virtual field experience. Examples of these comments include, "The stereonet tool is really cool! Stereonets in mineralogy were so confusing –this illustrates how they work spatially so well.", "I am 100% a visual learner... So what you are doing is right up my alley, and genuinely allowed me to learn more about structural geology, on a conceptual level, than any of my other previous classes."

**4 Cartoon-style Virtual Field Experience**

*Plate Tectonics and Materials of the Earth* is a first-year geology course at University of Washington that introduces rock
identification and geologic structures. The course traditionally culminates with an in-classroom mapping exercise for which
rock specimens are arranged around the lab room, and for which students produce a map, cross section, and geologic history.
By creating a fictional (but realistic) mapping exercise, we can adjust the complexity appropriate to introductory students and
create scenarios that draw on important concepts from the ten-week course. This type of classroom-based mapping exercise
also has the advantage of being accessible to many students (e.g., Cooke et al. 1999). During remote instruction for Winter
2021, we aimed to simulate this exercise within a virtual field experience. We produced a fictional field site, generally inspired
by the geology of northwestern Washington State (USA). We designed the map to include only a handful of different rock
types, with elements that represent accreted oceanic crust, folded terrestrial sedimentary rocks, arc volcanism, and simple
geologic structures with clear cross-cutting relationships.

The topography and lithology featured in the cartoon-style virtual field experience (Figs. 3a-b), were constructed in 3D from
a simple map and associated cross section, including a topographic profile based on relative resistance to weathering of each
rock type. The terrain model, outcrop surfaces, and various other game elements (e.g., trees, campfire, rock hammer, etc.) were
designed in Blender (Blender Foundation and Community 2020), a free and open-source 3D computer graphics software. To
make a hillshade-style base map of the terrain ("virtual lidar"), we rigged artificial lights in Blender to highlight the topography
of the terrain and then exported a gray-scale image of the shaded aerial view (Fig. 3c). Model outcrops began as cubes in
Blender and were subsequently bevelled to have varying bedding geometries. The 35 outcrops were positioned throughout the
terrain and geographically oriented; thus, if students identified the outcrop lithology and took the strike-and-dip of bedding,
they would be able to make the desired geologic map for the final project.

Identifying rocks at outcrops in the remote environment poses obvious problems of not being able to perform the standard
battery of tests on a physical specimen. To simulate examining rocks at outcrops, each outcrop had an adjacent rock-hammer
icon (Figs 3A and 3B), which opened an on-screen canvas with information about an outcrop. These informational canvases
included photographs (of real outcrops of the rock type, hand samples, and/or petrographic-microscope images), and brief
descriptions of characteristics that could not be conveyed from the photos alone.


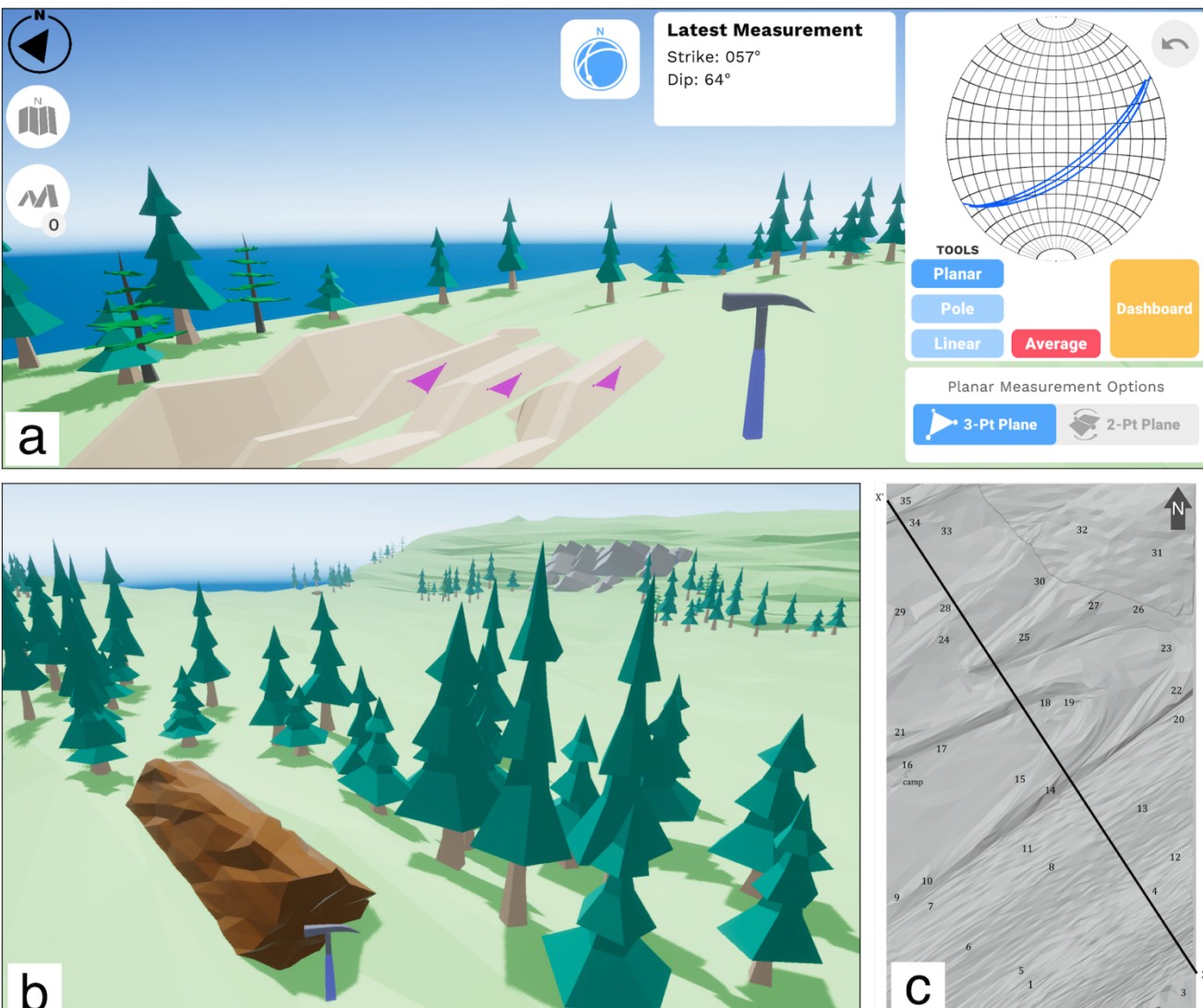

**Figure 3:** Fictional virtual field experience. (a) Planar measurements (magenta triangles) are taken on the bedding surfaces of a sandy-colored outcrop the with orientation data projected into a stereonet and printed onscreen. When the user clicks on the rock hammer, an informational canvas (not included in figure) opens to display photographs and other information about the outcrop. (b) A screenshot from a different part of the terrain shows two outcrops: a brown outcrop without bedding (foreground), and a blocky gray outcrop in the background. (c) The hillshade-style basemap provided to students includes the approximate locations (and numbers for organization) of outcrops and the position of X-X' cross section line for interpreting the subsurface during/after map construction.

The cartoon-style virtual field experience was hosted on itch.io (itch.io 2021), a website for hosting, selling, and downloading videogames. Since the virtual field experience is WebGL-based, students could explore the terrain in their internet browsers without having to permanently download files. Itch.io allows for user paywalls and/or for password protection of games. We password-protected the virtual field experience and provided a link and password within the assignment instructions. Students

connected to the game via their own computers (with various processor speeds and operating systems) and home internet connections (with various data rates). If student computers and/or internet connections could not support the game, the course

instructors were prepared to share their screens while using the virtual field experience as students directed the game-play; however, no issues of access were reported. Therefore, we assume that student computer systems that could successfully run common videoconference software (a requirement to participate in the class) were also sufficient for accessing the virtual field exercise. It is important to note, that some remote-learning situations in primary and secondary schools limit the types of websites students can visit; therefore, it may be important to consider such conditions depending on the intended audience of

a particular virtual field experience.

To complete the virtual field work of mapping, students worked in preassigned groups via videoconference, and much like in-person student field work, divided and alternated tasks. In video break-out rooms, one student shared their screen while moving through the terrain, measuring bedding orientations at outcrops, and activating the informational canvases. Other students used

the shaded terrain map to guide the main player, similar to how students use terrain maps during field mapping exercises. Other participants updated the group's shared spreadsheet to record bedding measurements and include objective descriptions, based on the informational canvases, for future discussions on rock identifications. With the orientation and lithologic data collected from the virtual field experience, students successfully generated geologic maps, and subsequently drafted cross sections and interpreted a geologic history for the fictional field site. Furthermore, students reported this assignment to be a highlight of the

online course, and several students mentioned showing the game to friends or family outside of class. These anecdotal reports of increased student engagement are corroborated by analytics that record the number of browser-plays between the time the assignment start date and due date: that number was 3-to-5 times greater than the minimum number required for student groups to complete the assignments. In other words, students visited the virtual field experience, on their own, for fun.

## 5 Final Remarks

The implementation of interactive virtual field experiences in the two courses successfully addressed the immediate need by substituting engaging online lessons for field and classroom exercises that were precluded by pandemic-related restrictions. Considering the stress and exhaustion that many students experienced as a result of the global pandemic, the students submitted work that conveyed comprehension of the intended educational goals. We also demonstrated that our structural-geology-query interface in Unity is successful for both structure-from-motion-generated models of actual field sites and custom-designed 3D

models of fictional geology. For the college courses, students did not report access issues with regards to where the virtual field experiences were hosted or how their personal computers and internet connections performed. That is, the games functioned well for more than 130 individual users across the globe, with an array of hardware, connection speeds, and browsers. With this contribution, we establish that these virtual field experiences are functional and accessible to many students and that students understand how to operate the tools and can make interpretations from the information they collect.

In response to the 2020 pandemic, the National Association of Geoscience Teachers and the International Association for Geoscience Diversity led a collaboration of more than 300 geoscience educators in developing a framework for designing

remote/virtual field experiences to meet the same learning outcomes as in-person exercises (Atchison et al. 2020). The exercises that we implemented within each virtual field experience address many of the highlighted learning outcomes (Table 1). Importantly, however, the open-ended nature of these simulations, and especially the Whaleback, enables other geoscience educators to design assignments tailored to different educational goals. We have made the Whaleback virtual field experience available to anyone (virtualfieldgeology.com 2021); at the time of this writing, hundreds of individuals have played the game,

including temporal clusters that suggest many instructors have designed their own field trips to this virtual site.

**Table 1:** Learning outcomes from the National Association of Geoscience Teachers and the International Association for Geoscience Diversity (Atchison et al. 2020) and the assignments associated with our two virtual field experiences.

| Learning Objectives (Atchison et al., 2020) | Whaleback Anticline — Examples of student tasks and sample assignment questions | Simplified fictional geology — Examples of student tasks and sample assignment questions |
|---|---|---|
| 1. Design a field strategy to collect or select data in order to answer a geologic question. | Strategized with the freedom of where to make observations and what data to collect. | Strategized how to most efficiently collect their data from a base map with numbered outcrop locations. |
| 2. Collect accurate and sufficient data on field relationships and record these using disciplinary conventions. | Measured the orientation of bedding, plotted map symbols and other data on orthophotos. | Made objective descriptions from photographs. Measured the orientation of bedding at outcrops. Plotted map symbols and contacts on maps. |
| 3. Synthesize geologic data and integrate with core concepts and skills into a cohesive spatial and temporal scientific interpretation. | Used stereonets to analyze variation in fold axis. Constructed a profile of fold from elevation data. | Constructed a geologic map. Constructed structural cross-section from the map. |
| 4. Interpret earth systems and past/current/future processes using multiple lines of spatially distributed evidence. | Searched for evidence to qualify whether the Whaleback's observable strain distribution is consistent with end-member kinematic fold models. | Wrote a step-by-step geologic history of the region from the stratigraphy, structures, and geochronological information. |
| 5. Develop an argument that is consistent with available evidence and uncertainty. | Q: Is the Whaleback a cylindrical fold? Q: Based on conceptual models of fold mechanics, infer the rock type at the core of the anticline | Q: Based on rock type and structure, are there potential sites for geologic carbon sequestration? |
| 6. Communicate clearly using written, verbal, and/or visual media with discipline-specific terminology appropriate to your audience. | Used graphics and text tools in shared, on-line documents to produce a project report | Submitted vector-designed geologic maps and cross-sections, and a written report of regional geologic history. |
| 7. Work effectively independently and collaboratively. | Worked independently with support from an online instructor and a community discussion board. | Collaborated on navigated, rock identification, and mapping through screen sharing. |


These virtual field experiences have a utility beyond the emergency transition to remote learning. Such simulations can continue to enable students to investigate 3D outcrops without the physical, geographic, and financial limitations often associated with field-based instruction. For example, the Whaleback Anticline is more than 4300 km from Seattle (an

impractical field trip), but we will continue to visit virtually, post-pandemic, via the game interface. We also see a role for these games in blended learning (c.f. Bond and Cawood 2021; this issue) and scientific visualization more broadly. In the virtual field experience, it is possible to observe and collect data from areas of the Bear Valley mine that are inaccessible on-foot; thus, the Whaleback game can augment in-person educational field trips to the site. Because the game is built on a high resolution, georeferenced terrain model, it could be used for research collaboration. Furthermore, the Whaleback is on private

land, and public/educational access is evolving. The immersive virtual field experience we have created can help to preserve a record of this geoheritage site (Geyer and Bolles 1979), should access be further restricted.

Benefits also persist for the virtual field experiences in fictional sites even after a full return to the classroom. While there are obvious advantages to the laboratory-based version of a fictional mapping exercise with real hand specimens for petrologic

description, the videogame-based exercise offers independent collection of structural data, and more information in the terrain. Our success in building a fictional mapping exercise suggests that geoscience educators need only imagination and bit of 3D modeling skills to design unlimited virtual geologic settings and structural histories for their students to map. It is thus possible to create idealized geology to permit students to discover foundational concepts. For example, imagine a unit on fold geometry in which students visit and measure bedding orientations on folds in the full range of shapes and attitudes, and can instantly

compare stereonets from each one. In these open-ended, fictional settings it is also possible to deliberately manipulate how much uncertainty the students encounter (c.f. Wilson et al. 2021, this issue).

The modular nature of the Unity game engine and our structural-geology-query tools means that it is relatively simple to build new virtual field experiences on other outcrop models. We intend to share the template and modules to enable others in the

geoscience community to generate virtual field experiences using our interface and their own terrain models. Future work includes creating instructions and workshop for this process, so that anyone with a 3D model can produce a field simulation without significant programming effort or experience. We see field simulations, like the two we describe here, as an emerging opportunity to provide the exploration of geologically interesting features without the typical limitations of field-based geology. Upon re-establishing many of our traditional practices as geologists (post-pandemic), we envision broader adoption

of videogame-based field experiences as one way to include more people in our geologic conversations.

**Author contributions**

MDN: conceptualization, methodology, visualization, software, project administration, and writing (original draft).  JGC: funding acquisition, supervision, methodology, writing (review & editing). JM: software, data curation; JFA: software.

**Competing interests**

The authors declare that they have no conflict of interest.

**Acknowledgements**

The authors acknowledge support from the US National Science Foundation (EAR-1523909 to JGC); and the University of Washington including the Royalty Research Fund, Department of Earth & Space Sciences, and UW Reality Lab. We thank our students for playing the game and sharing their impressions. We thank Mary Beth Gray and Arlo Weil for research

collaboration that led to the development of the Whaleback model, and Andrew Wang for contributions to an earlier version of the Whaleback game. We acknowledge Reading Anthracite Company for permission to photograph Bear Valley Strip Mine. The manuscript was improved by thoughtful comments from two anonymous referees and editor Steven Whitmeyer.

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
