# Peer review of "Virtual field experiences in a web-based videogame environment: Open-ended examples of existing and fictional field sites"

_Geoscience Communication, 2021_

## Author Response (AR1)

**EARTH & SPACE SCIENCES**

UNIVERSITY *of* WASHINGTON

College of the Environment

6 October 2021

Prof. Steven Whitmeyer
Editor, Special Issue
Geoscience Communication

Dear Dr. Whitmeyer:

Thank you for the thoughtful review of our manuscript for the GC Special Issue on virtual geoscience education resources. We have addressed the line-by-line comments of the two reviewers and indicated these changes in the right-hand margin of this revised manuscript.

In response to these reviews, we have expanded our discussion of student engagement with our virtual structural geology tools and how students responded to virtual field exercises. Because of the emergent nature of our project (emergency response to pandemic teaching restrictions), this is not a typical geoscience education study. In this manuscript our goal is to establish that our virtual field experiences are accessible to many students, the tools are functional, and that students can make structural interpretations from the information they collect. By sharing these tools, we hope to inspire future collaboration with geoscience education experts to further investigate the educational effectiveness of these VFEs.

To address the request for information about how the virtual field trips were developed, we provide a link to our Github page which includes detailed instructions on how educators can build virtual field excursions like ours using the software (also available on the Github page) that we developed. We have a second manuscript in preparation that will address the more technical aspects of the software and how our "structural geology query toolkit" was developed.

We hope that these two additions of student engagement and a link to our workflow instructions and software enhance the value of this manuscript to the community. Thank you for your consideration.

Sincerely,

Mattathias (Max) Needle
Ph.D. candidate
University of Washington
mneedle@uw.edu

Box 351310 ● 070J Johnson Hall ● Seattle, WA 98195-1310

PHONE (206) 543-8715   FAX (206) 543-0489 ● www.ess.washington.edu

**Overview and General Recommendation:**

Needle et al. describe two new and exciting virtual geology field experiences and share one of them with the larger earth science community. However, the author's do not present any geologic or education data or results within this contribution, thus I do not feel this contribution qualifies as a Research Article per the Geoscience Communication Review Criteria. The contribution primarily serves as a report on the sites they created and how they chose to implement them in the classroom. However, it is an important resource contribution to the geoscience community and should be published.

The field trips include one real world site, the Whaleback anticline in Central Pennsylvania, recreated digitally using structure-from-motion photogrammetry and one fictional location created using Blender, an open-source 3D creation platform. Both field sites were designed in Unity, a platform that allows the user to build 2D, 3D, and VR games. The Whaleback anticline field site "video game" is publicly available for community use on a free video-game hosting platform – web links are available via the website virtualfieldgeology.com, which is run by the authors.

These are useful tools to supplement, replace, and/or enhance traditional geoscience field education. Virtual field trips (VFTs) provide improved access for students and may ultimately alleviate some barriers associated with field excursions. I suspect geoscience instructors will greatly benefit from using the two VFTs created and described by the authors. The authors leave the goals and activities associated with the VFTs open-ended to allow other instructors to make use of the field sites as they see fit. I recommend the manuscript be published with minor revision and with recognition that it is not a research article.

The contribution has two primary weaknesses. First, the authors do not present any quantitative or evaluative data from their pilot VFT implementations at the University of Washington. As such I cannot assess 1) student engagement, 2) if students acquired and accomplished the associated field skills and learning goals, or 3) how student learning gains compare to traditional field experiences. I recognize that this falls into the category of discipline-based education research with human research subjects. As such, to collect and report much of the data I request the authors would need Institutional Review Board approval. I encourage the authors to carefully consider how to assess the success of their VFTs in the future, I'd be very interested in those results!

Second, the authors do not provide sufficient workflow or technical information regarding how they built their VFTs to be useful for others to follow their approach. The manuscript would benefit from a workflow figure and detailed steps for others to create VFTs, as this seems to be the primary purpose of their contribution. At present the manuscript is simply a description of the sites they built.

The authors have provided neat tools for the community, but their contribution to GC serves only to share and describe the tools they created as opposed to presenting information and data assessing the tools or a more detailed approach to building the tools. I think both would substantially improve the utility of the contribution.

**Specific Comments:**

Lines 58-59: The manuscript would benefit from additional references of geoscience education research to support several claims. References here would help.

Line 120: Do you have user results from the July 2020 pilot study to report?

Line 139: How do you measure user experience such as "Most students found the game controls to be intuitive"?

Line 216: Do you have any user results beyond anecdotal accounts?

Line 224: Can you elaborate on high-quality work? How was this assessed?

Lines 258-264: I only caution the authors to consider that fictional environments could be abused by instructors. If a major goal of using VFTs is to accessibly acquire and practice field skills, then the VFTs should be geologically reasonable and possible.

Line 266: I look forward to this template contribution – this will be excellent for the geoscience community!
* * *
**Authors' response to anonymous referee #1**

Thank you very much for the helpful and positive comments. We agree the results of a carefully designed study (with IRB approval) on the effectiveness of the virtual field trips would be very interesting! An exciting aspect of the tools we introduce here is the possibility for direct comparison of student learning outcomes after virtual versus actual field experiences at classic localities like the Whaleback Anticline, with the same, open-ended and data-oriented questions. We look forward to opportunities to collaborate with science education experts on such studies in the future. Here we report on pandemic-related "emergency" replacements for field exercises, with a prototype game to provide proof-of-concept for our approach. We show that these VFTs are functional and accessible to many students and that students understand how to operate the tools and can make interpretations from the information they collect. In response to your comments, we have expanded our discussion of student engagement with these tools, and include some representative anecdotal responses.
We intend to share our workflow for building these VFTs in a forthcoming manuscript but are eager to have these examples as part of this special issue.

Lines 58-59: The manuscript would benefit from additional references of geoscience education research to support several claims. References here would help.

--Corrected. In the revised manuscript, the claims in these two sentences have been removed. Thanks!

Line 120: Do you have user results from the July 2020 pilot study to report?

Line 139: How do you measure user experience such as "Most students found the game controls to be intuitive"?

Line 216: Do you have any user results beyond anecdotal accounts?

--In the revised manuscript we provide further discussion of the collected student responses to our work to date.

Line 224: Can you elaborate on high-quality work? How was this assessed?

--Line revised.  Student work was compared to our recollection of student work in similar exercises from prior (in-person) offering of the course.

Lines 258-264: I only caution the authors to consider that fictional environments could be abused by instructors. If a major goal of using VFTs is to accessibly acquire and practice field skills, then the VFTs should be geologically reasonable and possible.

--Agreed!
* * *
**Anonymous referee #2, 23 July 2021**

**General Comments**

Needle et al. present an exciting contribution to geoscience education with new virtual field sites for a range of students. I thoroughly enjoyed reading this paper and can imagine using it in my own classroom, however, without any data presented regarding student feedback or pedagogical theory, it needs major revision before publication as a research article. As it stands now, it is more an article on how the game faired, not a study on its impact or use.

Mainly, the authors do not present any quantitative feedback from students on games feasibility. Words used to assess the game were relative and speculative. With the addition of quantitative data relating to student use and feedback, this paper has the potential to fully address relevant scientific questions within the scope of GC. This is a significant classroom advancement with the potential to be implemented immediately but there is little to no inquiry as to how to build the tool in other classrooms. This manuscript would be benefitted by 1) more data involving student feedback, 2) more data on the type of computers used, 3) more information on how to build the environments. The paper was well written and had few technical errors. It was exciting to read and I look forward to reading the final manuscript

**Technical Corrections:**

Line 30 - I agree with your statement that "The need for educational experiences that incorporate fundamentals will remain after the pandemic.", but evidence is needed. It is a strong statement to write without any examples as to why.

Line 35 – I would argue that it is closely tied still

Line 50 – Erase the "," after "virtual field experience"

Line 72 - Does your line take in to account elevation changes?

Line 108 – "with steeply-dipping/overturned limbs"

Line 91 – you have already defined SfM, you can use it from here on out

Line 119 – I think this aerial image would be a great addition to figure 1

Line 139 – This is great, but where is the data? How many is "most"? I'd like to see the student feedback associated with this exercise

Line 140 – What type of machines were they using, with how much RAM and which type of graphic card? Have you tested it on computers with less RAM or an unstable wifi connection?

Figure 2 – Very cool graphics! Is there any scale available?

203 – What type of accommodations were prepared?

228 – I think this is fantastic news but would like information on the "array of hardware, connection speeds, and browsers" for guidance when other professors implement this game.
* * *
**Authors' response to Anonymous referee #2**

Thank you very much for the helpful and positive comments. Our goal in this manuscript is to show that these VFTs are functional and accessible to many students, that students understand how to operate the tools and that they can make interpretations from the information they collect. In response to your comments, we have expanded our discussion of student engagement with these tools, and include some representative anecdotal responses. Although we have not collected information about which computers students have used to access the virtual field experiences, we do know that anyone who can successfully participate in class via common video-conference software could play the games -- and we add this information to the manuscript. We intend to share our workflow for building these VFTs in a forthcoming manuscript, but are eager to have these pilot examples as part of this special issue on online education, as a demonstration of new possibilities for online and hybrid geoscience education.

Line 30 - I agree with your statement that "The need for educational experiences that incorporate fundamentals will remain after the pandemic.", but evidence is needed. It is a strong statement to write without any examples as to why.

--Agreed! We elaborate in our final remarks.

Line 35 – I would argue that it is closely tied still

--Yes, as we note on lines 37-40.

Line 50 – Erase the "," after "virtual field experience"

--Corrected. Thanks.

Line 72 - Does your line take into account elevation changes?

--Yes! Clarified. Thanks.

Line 108 – "with steeply-dipping/overturned limbs"

--Corrected. Thank you.

Line 91 – you have already defined SfM, you can use it from here on out

--Right.  Thanks.

Line 119 – I think this aerial image would be a great addition to figure 1

--It's part of Figure 2, and we have now noted that.

Line 139 – This is great, but where is the data? How many is "most"? I'd like to see the student feedback associated with this exercise

--We have added a new paragraph after line 165 with more information about the student response to the pilot exercise.

Line 140 – What type of machines were they using, with how much RAM and which type of graphic card? Have you tested it on computers with less RAM or an unstable wifi connection?

--We do not have information about the specific hardware that each student used, but given that none of our students reported difficulties, we expect that the requirements are no greater than that required to participate in a class over Zoom.

Figure 2 – Very cool graphics! Is there any scale available?

--Scale added.  Thanks!

203 – What type of accommodations were prepared?

--Clarified in the text.

228 – I think this is fantastic news but would like information on the "array of hardware, connection speeds, and browsers" for guidance when other professors implement this game.

--Please see our response to the comment on line 140 (above).